# EPIPOLAR GEOMETRY IMPROVES VIDEO GENERATION MODELS

## ABSTRACT

Video generation models have progressed tremendously through large latent diffusion transformers trained with rectified flow techniques. Yet these models still struggle with geometric inconsistencies, unstable motion, and visual artifacts that break the illusion of realistic 3D scenes. 3D-consistent video generation could significantly impact numerous downstream applications in generation and reconstruction tasks. We explore how epipolar geometry constraints improve modern video diffusion models. Despite massive training data, these models fail to capture fundamental geometric principles underlying visual content. We align diffusion models using pairwise epipolar geometry constraints via preference-based optimization, directly addressing unstable camera trajectories and geometric artifacts through mathematically principled geometric enforcement. Our approach efficiently enforces geometric principles without requiring end-to-end differentiability. Evaluation demonstrates that classical geometric constraints provide more stable optimization signals than modern learned metrics, which produce noisy training signals. Training on static scenes with dynamic cameras ensures metric quality while the model still generalize to various dynamic scenes. By bridging data-driven learning with classical geometric computer vision, we present a practical method for generating 3D consistent videos without compromising visual quality.

## 1 INTRODUCTION

Video generation has witnessed remarkable progress, with recent models (OpenAI, 2024; Wiedemer et al., 2025; Polyak et al., 2025; Wang et al., 2025a; Kong et al., 2024) producing increasingly realistic content from text and image conditions. This advancement has spurred researchers to repurpose these powerful video models for broader applications, including animation (Yang et al., 2024), virtual worlds generation (He et al., 2025), and novel view synthesis (Zhou et al., 2025). Video diffusion models are trained on vast volumes of data, developing strong understanding of object appearance, motion patterns, and scene composition. Many recent works aim to utilize these priors in various downstream tasks (Jiang et al., 2025; Voleti et al., 2024; Chen et al., 2024). Despite this, these models still struggle to maintain perfect 3D consistency, often producing content with imperfect geometry, unstable motion, and perspective flaws, even though almost all training data is 3D consistent. Some approaches for enhancing 3D consistency rely on noise optimization (Liu & Vahdat, 2025), explicit guidance through point clouds (Zhang et al., 2024; Hou et al., 2024), or camera parameters (Zheng et al., 2024). Nevertheless, noisy control signals can constrain the model's generation capabilities, and the latent space optimization makes it difficult to compute direct geometric losses.

With the rising popularity of reinforcement learning for model alignment (Rafailov et al., 2023; Shao et al., 2024; Ouyang et al., 2022), post-training alignment has gained attention in diffusion model research. Methods such as VideoReward (Liu et al., 2025) finetune vision-language models on human preference data, enabling direct supervision through reward models. However, human-annotated quality scores introduce noisy signals and are expensive to collect. Human judgments are inherently subjective and may not capture geometric principles ensuring 3D consistency. The gap between subjective human evaluations and objective geometric requirements creates an opportunity for alignment methods that leverage more mathematically grounded metrics for video quality assessment.

We propose a simple approach that bridges modern video diffusion models with classical computer vision algorithms. Rather than incorporating explicit 3D guidance during generation, we use well-

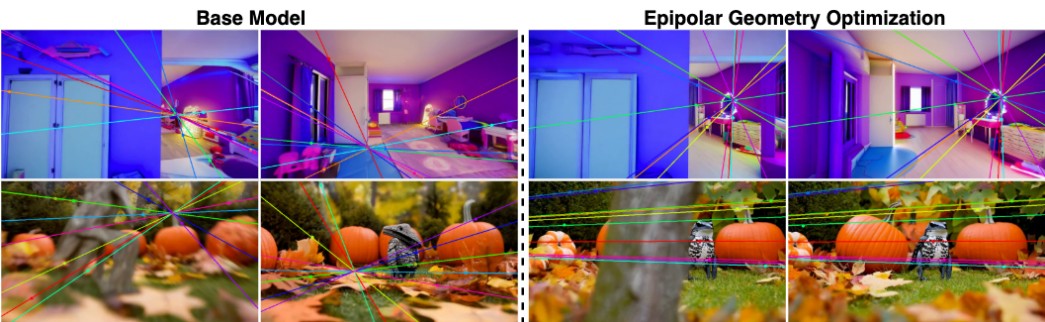

Figure 1: First and middle frame from videos. The baseline model produces geometrically inconsistent outputs with artifacts and unnatural motion. Aligned model generates visibly improved results with smoother camera trajectories, reduced artifacts, and enhanced 3D consistency.

established non-differentiable geometric constraints as reward signals in a preference-based finetuning framework. Specifically, we leverage epipolar geometry constraints to assess 3D consistency between frames. By sampling multiple videos conditioned on the same prompt, we generate diverse camera trajectories that vary in geometric coherence. Epipolar geometry metrics provide reliable signals for identifying which generations better adhere to projective geometry principles, enabling us to rank videos and create training pairs that guide the model toward improved geometric consistency.

Our method implements this through Direct Preference Optimization (DPO) (Rafailov et al., 2023), requiring only relative rankings rather than absolute reward values. This bypasses the difficulties of directly using non-differentiable computer vision algorithms in the training loop. By finetuning the model to prioritize generations that satisfy classical geometric constraints, we guide it towards generating inherently more 3D-consistent videos without restricting creative capabilities or requiring explicit 3D supervision. As shown in Figure 1, this results in enhanced 3D consistency, smoother camera trajectories, and fewer artifacts compared to the baseline model.

While simple in nature, this paper shows that a basic geometric constraint, described in 1982 (Sampson, 1982), can recover what video models fail to do, even after large-scale training on billion-scale data: 3D consistency.

In summary, the key contributions are as follows:

**Epipolar Geometry Optimization:** We introduce a method for finetuning video diffusion models using epipolar geometry constraints as reward signals, particularly leveraging Sampson distance to enhance 3D video consistency without needing differentiability. The models finetuned with simple yet reliable signals from classical vision algorithms achieve superior consistency and quality, significantly reducing artifacts and unstable motion in generated content. Our approach demonstrates that aligning models with fundamental geometric principles enables the generalizable shaping of the model's sample distribution, which is independent of the video content and generalizes to dynamic content.

**Comprehensive Evaluation Framework:** We develop an extensive evaluation protocol measuring perceptual quality, 3D consistency, motion stability, visual fidelity, and generalization across diverse scenarios. We also compare multiple alignment approaches, demonstrating that classical geometric constraints provide more stable optimization signals than modern learned metrics.

**Large-Scale Preference Dataset:** We create and release a large dataset of over 162,000 generated videos annotated with 3D scene consistency metrics, enabling further research in geometry-aware video generation. In addition, we generate a large library of captions with amplified camera motion and a data mining pipeline allowing to reuse the approach for other models and subtasks. The data includes diverse prompts spanning natural landscapes, architectural scenes, and dynamic environments, each with multiple video generations.

## 2 RELATED WORK

We structure the related work section into generative models and post-training methods to adapt them.

## 2.1 Video Generation Models

Recent advances in video generation have been dominated by closed-source models developed by well-resourced technology companies. These models, trained on large proprietary datasets with computational resources beyond academic reach, have demonstrated remarkable capabilities while revealing limited architectural details. Notable releases include OpenAI's Sora (OpenAI, 2024), Runway's Gen-2 and Gen-3 (Runway, 2024), Luma AI (LumaLabs, 2024), Pika Labs (PikaLabs, 2024), and Google DeepMind's Veo series (Google DeepMind, 2024). While these systems produce impressive results, their closed nature limits opportunities for finetuning or adaptation to other vision tasks. Open-source large latent diffusion models have recently become available, increasing interest in improving video generators. Stable Video Diffusion (Blattmann et al., 2023) developed efficient training strategies, Hunyan-Video (Kong et al., 2024) presented systematic scaling approaches, LTX-Video (HaCohen et al., 2024) introduced real-time optimizations, and Wan-2.1 (Wang et al., 2025a) introduced an efficient 3D Variational Autoencoder with expanded training pipelines. Wan-2.1 offers 1.3B and 14B parameter versions, enabling researchers to explore adaptation techniques for various downstream tasks. These models are trained on enormous data volumes covering more content variety than specific applications need, making domain-aware alignment valuable. V3D (Chen et al., 2024) finetunes models for 3D reconstruction, while VideoReward (Liu et al., 2025) introduced reinforcement learning-based alignment. However, prior methods rely on subjective human preferences or vision language models trained to mimic them. Our approach optimizes against mathematical rules from epipolar geometry, providing clean signals that align models with fundamental 3D consistency principles rather than subjective judgments.

## 2.2 Diffusion Models Alignment

Since image and video latent diffusion models are trained on internet-scale noisy data, efficient finetuning and alignment strategies have emerged as active research areas. Latent image diffusion models (Podell et al., 2023; Rombach et al., 2022) finetune models on data highly ranked by aesthetics classifiers (Schuhmann, 2022). DRAFT (Clark et al., 2023) and AlignProp (Prabhudesai et al., 2023) explore this paradigm by tuning diffusion models to maximize reward functions directly. DPOK (Fan et al., 2023) and DDPO (Black et al., 2023a) expand the paradigm to introduce distributional constraints. Diffusion-DPO (Wallace et al., 2024) introduces Direct Preference Optimization into diffusion model alignment. In contrast to other approaches, DPO does not require direct access to reward models and can be trained with only pairwise preference data. Additionally, this eliminates the need to decode final denoised samples, enabling finetuning directly in latent space and significantly improving training efficiency. Recently, VideoReward (Liu et al., 2025) adapted Diffusion-DPO for video alignment, effectively aligning video generation with human preferences. Yet all these approaches focus on optimizing for subjective and noisy human evaluation. Lately, DSO (Li et al., 2025b) employs DPO to align 3D generators with physical soundness, and PISA (Li et al., 2025a) improves physical stability of video generators with multi-component reward functions. Our method leverages classical computer vision algorithms to provide objective, mathematically grounded preference signals based on epipolar geometry, resulting in more reliable and consistent alignment with 3D physical principles than approaches relying on learned or subjective metrics.

## 3 Method

We aim to align pretrained video diffusion models to generate geometrically consistent 3D scenes from text or image prompts. We propose an alignment strategy leveraging classical epipolar geometry constraints within a preference-based optimization framework. Traditional reinforcement learning methods (Black et al., 2023b; Shao et al., 2024) require explicit reward functions and access to final samples, which is impractical for video models due to lack of differentiable reward models and high denoising computational costs. Our key observation is that while classical epipolar geometry constraints do not produce smooth, globally comparable loss surfaces across different scene types, the relative intra-prompt error remains consistent. When generating multiple video sequences with fixed conditioning, diffusion sampling's stochastic nature produces outputs with varying geometric consistency degrees. Epipolar error effectively quantifies relative 3D consistency, with higher values indicating lower geometric consistency. This finding aligns with the direct preference optimization (DPO) paradigm, which requires only relative metrics to determine preference between output pairs

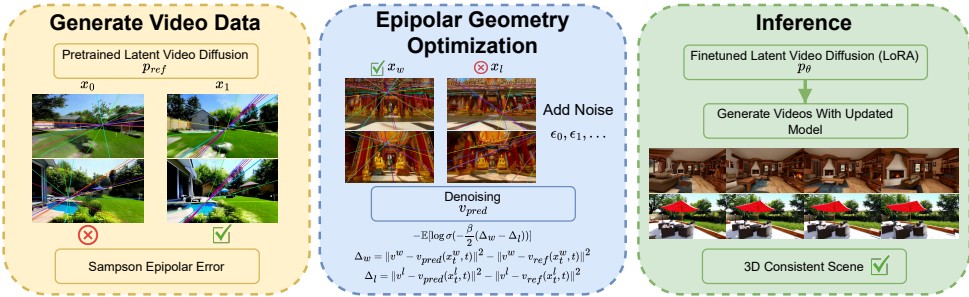

Figure 2: **Epipolar Geometry Optimization pipeline.** Our approach: (1) Generate diverse videos using pretrained generators (Wang et al., 2025a) and leverage the Sampson epipolar error to identify 3D consistent vs. inconsistent samples; (2) Train policy $p_\theta$ using Flow-DPO (Liu et al., 2025) with the static penalty to prefer geometrically consistent outputs; (3) Apply the updated policy to enhance 3D consistency in the base video diffusion model.

rather than absolute reward values. DPO's pairwise comparison nature eliminates the need for globally normalized reward functions, instead leveraging reliable local ranking provided by epipolar geometry measurements to guide model alignment toward more geometrically consistent video generation.

### 3.1 OBJECTIVE FUNCTION

Given the pretrained video generator $p_{\text{ref}}$ that takes a text prompt and optional first frame conditioning $I$ and generates video samples $x_0 \sim p_{\text{ref}}(x_0|T, I^*)$, where $I^* \in \{I, \emptyset\}$ we want to learn model $p_\theta$ optimized to generate 3D-consistent video sequences. One approach would be to optimize:

$$\max_\theta \mathbb{E}_{(T, I^* \in \{I, \emptyset\}) \sim \mathcal{D}_c, x_0 \sim p_\theta(x_0|T, I^*)} [r(x_0)]$$
$$- \beta \mathbb{D}_{\text{KL}} [p_\theta(x_0|T, I^*) \| p_{\text{ref}}(x_0|T, I^*)], \tag{1}$$

where $r(x_0)$ outputs 3D consistency scores. However, this formulation presents critical challenges: the reward function relies on non-differentiable classical computer vision algorithms, and requires complete video generation for evaluation, making traditional reinforcement learning impractical. This motivates our adoption of DPO (Rafailov et al., 2023; Wallace et al., 2024).

Given dataset $\mathcal{D}(\{c, x_0^w, x_0^l\})$ with condition $c$ and sample pairs from $p_{\text{ref}}$ where $x_0^w$ has higher reward than $x_0^l$ ($x_0^w \succ x_0^l$), Diffusion-DPO (Wallace et al., 2024) solves eq. (1) analytically. For rectified flow models (Lipman et al., 2022; Liu et al., 2022; Albergo & Vanden-Eijnden, 2022), the Flow-DPO loss (Liu et al., 2025) is:

$$\mathcal{L} = -\mathbb{E} \Bigg[ \log sigmoid \Bigg( -\frac{\beta_t}{2} \Big( \|v^w - v_\theta(\mathbf{x}_t^w, t)\|^2 - \|v^w - v_{\text{ref}}(\mathbf{x}_t^w, t)\|^2$$
$$- \big( \|v^l - v_\theta(\mathbf{x}_t^l, t)\|^2 - \|v^l - v_{\text{ref}}(\mathbf{x}_t^l, t)\|^2 \big) \Big) \Bigg) \Bigg], \tag{2}$$

where $\beta_t = \beta(1 - t^2)$ and $x_t^* = (1 - t)x_0^* + t\epsilon^*$.

To prevent degenerate solutions where the model reduces motion to achieve 3D consistency, we add a temporal variation penalty:

$$\mathcal{L}_{\text{temporal}} = -\lambda \cdot \mathbb{E}[\text{Var}_t(\hat{x}_0)] \tag{3}$$

where $\hat{x}_0 = x_t + (1 - t) \cdot v_\theta(x_t, t)$ is the predicted clean sample, variance is computed across the temporal dimension, and $\lambda = 0.001$. Our final objective combines both terms: $\mathcal{L}_{\text{total}} = \mathcal{L} + \mathcal{L}_{\text{temporal}}$.

Minimizing this loss encourages the model to improve denoising performance on preferred samples $\mathbf{x}_t^w$ relative to less preferred samples $\mathbf{x}_t^l$, guiding the predicted velocity field $v_\theta$ to align with videos exhibiting better 3D consistency while preserving motion quality.

## 3.2 3D Consistency Metric

We evaluate the 3D consistency of generated videos by validating how well they satisfy epipolar geometry constraints. Epipolar geometry represents the intrinsic projective relationship between two views of the same scene, depending only on the camera's internal parameters and relative positions. In perfectly consistent 3D scenes, corresponding points across different viewpoints must adhere to these geometric constraints.

For any two corresponding points $\mathbf{x}$ in one frame and $\mathbf{x}'$ in another, the epipolar constraint $\mathbf{x}'^T \mathbf{F} \mathbf{x} = 0$ must be satisfied, where $\mathbf{F}$ is the fundamental matrix. This constraint ensures that a point in one view must lie on its corresponding epipolar line in the other view. The fundamental matrix encapsulates the geometric relationship between the two camera poses. It can be formulated as $\mathbf{F} = [\mathbf{e}']_\times \mathbf{P}' \mathbf{P}^+$, where $\mathbf{P}$ and $\mathbf{P}'$ are the camera projection matrices, $\mathbf{P}^+$ is the pseudo-inverse of $\mathbf{P}$, and $\mathbf{e}'$ is the epipole in the second view.

Given a pair of frames $\mathbf{x}_i$ and $\mathbf{x}_j$ from a generated video, we first compute a set of point correspondences using SIFT (Lowe, 1999) feature matching. While we validate the method with a simple, robust handcrafted descriptor, the pipeline can also leverage more recent learned descriptors (Lindenberger et al., 2023; Sun et al., 2021; Potje et al., 2024). These correspondences provide a robust set of matching points between the different viewpoints. We then estimate the fundamental matrix using the normalized 8-point algorithm within a RANSAC (Fischler) framework to handle outliers.

Once we have estimated the fundamental matrix, we can measure the geometric consistency using the Sampson epipolar error (Sampson, 1982):

$$S_E = \frac{(\mathbf{x}'^T \mathbf{F} \mathbf{x})^2}{(\mathbf{F}\mathbf{x})_1^2 + (\mathbf{F}\mathbf{x})_2^2 + (\mathbf{F}^T\mathbf{x}')_1^2 + (\mathbf{F}^T\mathbf{x}')_2^2} \tag{4}$$

The Sampson error provides a first-order approximation to the geometric distance between a point and its epipolar line. Lower Sampson error values indicate better adherence to projective geometry constraints and, thus, more consistent 3D structure in the generated videos.

## 3.3 Implementation Details

We conduct experiments with a state-of-the-art open-source video diffusion model Wan2.1 (Wang et al., 2025a), with 1.3 billion parameters.

**Offline Dataset Generation:** Since our method focuses on 3D-consistent scene generation, we require videos of static scenes with dynamic camera movements. We extract text prompts from the DL3DV (Ling et al., 2024) and RealEstate10K (Zhou et al., 2018) datasets, provided by (Zheng et al., 2025), containing a wide variety of indoor and outdoor scenes. We deliberately selected these datasets because they feature dynamic cameras in static scenes, where epipolar constraints are valid. Dynamic objects would corrupt geometric measurements by violating single-camera assumptions. This is precisely one of our key insights: training on captions describing dynamic cameras in static scenes ensures high metric quality. Since the motion is encoded in early iterations (Wu et al., 2024), a combination of implicit KL-divergence regularization with the reference model and a static content penalty allows for the encoding of generalizable modifications to the model's sample distribution that are independent of the video content.

To enhance training data quality, we employ Gemma-3 VLM to expand original prompts with more challenging camera motion descriptions, increasing geometric complexity and ensuring sufficient variance in 3D consistency. We generate three videos per caption to increase variation in 3D consistency quality, as preliminary experiments showed pairs from just two samples often lacked meaningful geometric differences. We implement rigorous data filtering: in addition to removing near-static videos, we only sample pairs where $(\text{metric}(x_{\text{win}}) - \text{metric}(x_{\text{lose}}) > \tau) \wedge (\text{metric}(x_{\text{win}}) > \epsilon)$, eliminating pairs where both videos have similar consistency and ensuring we only learn from meaningful gaps. In total, we generate 24,000 triplets for text-to-video and 30,000 triplets for image-to-video training, requiring approximately 1,980 GPU hours on NVIDIA A6000s.

**Training Configuration:** Given the computational cost of fine-tuning video diffusion models, we implement our approach using Low-Rank Adaptation (LoRA) (Hu et al., 2022) with rank $r = 64$ and $\alpha = 128$. This strategy eliminates the need to store the reference model in memory, since the base

model with the adapter disabled naturally serves as $p_{\text{ref}}$ during training. We train with a batch size of 32 for 10,000 iterations using the AdamW (Loshchilov & Hutter, 2017) optimizer with a learning rate of $5 \times 10^{-6}$ and 500 warmup steps. The finetuning takes 2 days on 4 A6000 GPUs.

# 4 EXPERIMENTS

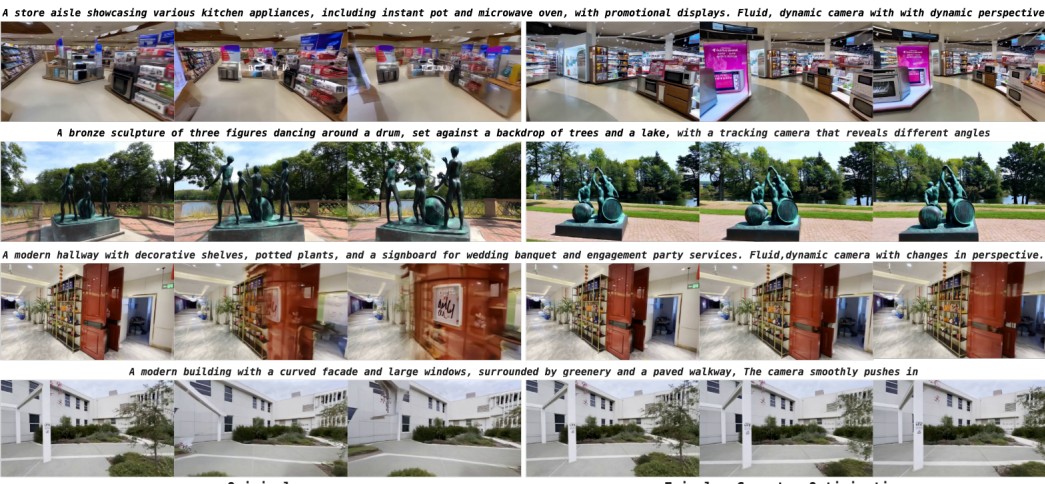

Figure 3: **Qualitative Evaluation:** Visual comparison between the videos generated by the base and finetuned model. First two rows: Wan-2.1-T2V (Wang et al., 2025a), Last two: Wan-2.1-I2V. Our finetuning significantly reduces artifacts and enhances motion smoothness, resulting in more geometrically consistent 3D scenes. Best seen in the supplementary video.

## 4.1 EVALUATION SETUP

We evaluate epipolar-aligned model across three dimensions: 3D consistency, motion stability, and generalization to dynamic scenes. We demonstrate that classical geometric constraints provide more reliable optimization signals than learned metrics while improving video generation quality.

**Data and Metrics:** We evaluate on 400 videos from DL3DV (Ling et al., 2024) and RealEstate10K (Zhou et al., 2018) test sets, using Gemma-3 VLM (Team et al., 2025) to generate challenging camera motion descriptions. For generalization, we test on VBench 2.0 (Huang et al., 2024), MiraData (Ju et al., 2024) and VideoReward (Liu et al., 2025) benchmarks extending beyond static scenes. We measure performance using: (1) VideoReward VLM for motion quality assessment, (2) VBench protocol (Huang et al., 2024) for standardized motion and visual quality metrics, (3) classical geometric consistency via Sampson epipolar error, and (4) 3D reconstruction quality via Gaussian Splatting to validate downstream task impact.

**Human Evaluation:** We conduct two-stage human evaluation. First, annotators label videos as geometrically consistent based on visible artifacts and motion stability. This reveals that the baseline produces consistent videos only 54.1% of the time, confirming significant room for improvement despite the model's strong capabilities. Second, annotators perform pairwise comparisons between baseline and aligned versions. This protocol demonstrates that our approach preserves quality for already-consistent content while dramatically improving inconsistent cases (60.4% vs 7.5% win rate).

## 4.2 3D CONSISTENCY

We validate that epipolar geometry alignment improves 3D consistency using three approaches that test different aspects of geometric quality.

**3D Scene Reconstruction:** We test whether generated videos support accurate 3D scene reconstruction using VGGT (Wang et al., 2025b) to extract scene parameters and camera trajectories. We

Table 1: **3D Consistency Evaluation:** Epipolar aligned model improves 3D Scene Reconstruction and is preferred by human evaluators.

| Method | 3D Consistency Metrics | | 3D Scene Reconstruction | | | Human Eval |
|---|---|---|---|---|---|---|
| | Sampson Error ↓ | Perspective Realism ↑ | PSNR ↑ | SSIM ↑ | LPIPS ↓ | Consistency Rate |
| Baseline | 0.190 | 0.426 | 22.32 | 0.706 | 0.343 | 54.1% |
| Ours | **0.131** | **0.428** | **23.13** | **0.729** | **0.315** | **71.8%** |

initialize 3D Gaussian Splatting from extracted scene structure, run 7000 optimization iterations using Splatfacto (Tancik et al., 2023) on 80% of frames, and evaluate reconstruction fidelity on the remaining 20%. Our model demonstrates substantial improvements: PSNR increases from 22.32 to 23.13 (+3.6%), SSIM improves from 0.706 to 0.729 (+3.2%), and LPIPS decreases from 0.343 to 0.315 (-8.2%). These gains demonstrate that epipolar alignment produces videos with genuinely enhanced 3D structure rather than superficial improvements.

**Geometric Consistency Metrics:** We directly measure adherence to projective geometry principles using classical computer vision algorithms. The Sampson epipolar error shows a dramatic 31% reduction from 0.190 to 0.131, verifying that our alignment successfully optimizes the metric used for preference selection and confirming that classical epipolar geometry provides clean optimization signals. Additionally, perspective realism, measured by a model trained to evaluate whether image frames contain realistic perspective (Sarkar et al., 2024) improves from 0.426 to 0.428, demonstrating positive impact on adjacent geometric metrics despite this metric's inherent noise.

**Human Evaluation:** While numerical metrics capture geometric aspects, 3D inconsistencies often manifest as artifacts, jitter, or unnatural changes that humans excel at detecting because they make scenes appear unrealistic. Annotators evaluated videos for scene consistency, realism, and artifact-free content. Our method generates significantly more plausible scenes, with 71.8% of videos labeled as geometrically consistent compared to only 54.1% for baseline content. This 17.7 percentage point improvement demonstrates that geometric alignment benefits are apparent to human observers.

### 4.3 MOTION QUALITY

Table 2: **Motion Quality Evaluation:** Epipolar aligned model improves motion stability and is preferred by human evaluators despite dynamics-consistency tradeoffs.

| Method | VBench Motion Metrics | | | VideoReward | Motion Level | Human Eval |
|---|---|---|---|---|---|---|
| | Motion Smoothness ↑ | Dynamic Degree ↓ | Temporal Flickering ↑ | Motion Quality ↑ | Mean SSIM ↓ | Motion Preference Rate |
| Baseline | 0.981 | **0.751** | 0.958 | 50.0% | 0.233 | 18.5% |
| Ours | **0.984** | 0.710 | **0.969** | **69.5%** | **0.223** | **53.2%** |

While geometric alignment enforces smooth, consistent motion and reduced jitter, we also verify that our alignment preserves the model's ability to generate diverse motions. We evaluate motion quality using various metrics, focusing on different aspects, including temporal dynamics, perceptual assessment, and human preference.

**Temporal Dynamics:** VBench motion metrics show mixed results that reflect the dynamics-consistency tradeoff. Motion smoothness and temporal stability improves, indicating more stable frame-to-frame transitions. However, dynamic degree decreases from 0.751 to 0.710, suggesting reduced motion amplitude. Mean SSIM between first and remaining frames decreases as well, confirming model's ability to generate dynamic scenes. While we acknowledge dynamic-consistency tradeoff, single neural network metrics can exhibit bias, motivating the multi-metric evaluation.

**Perceptual Quality Assessment:** VideoReward motion quality evaluation shows substantial improvement with our method achieving 69.5% win rate compared to baseline. This human-distilled assessment validates that geometric consistency training produces motion that aligns better with human preferences for natural, stable video dynamics, despite some reduction in motion amplitude.

**Human Preference:** Direct human evaluation further proves the motion quality of our method, with annotators preferring our approach at 53.2% rate across all video types. Since annotators focus

attention on jitter and motion artifacts, our high preference rate showing that stability improvements outweigh motion amplitude reductions. This preference is particularly strong in initially inconsistent videos, where our method achieves 60.4% preference compared to just 7.5% for baseline.

## 4.4 GENERALIZATION

Table 3: **Generalization to Dynamic Scenes:** Despite training on static scenes only the model generalize well to various dynamic scenes showcasing the effectiveness of enforcing geometrical constraints.

| Benchmark | Visual Quality | Motion Quality | Text Alignment | Overall |
|-----------|----------------|----------------|----------------|---------|
| VBench 2.0 | 61.3% | 55.3% | 52.0% | 57.9% |
| VideoReward | 65.0% | 58.5% | 50.5% | 58.5% |
| MiraData | 57.0% | 58.0% | 52.0% | 58.5% |

| Method | Background Consistency | Aesthetic Quality | Temporal Flickering | Motion Smoothness | Dynamic Degree |
|--------|------------------------|-------------------|---------------------|-------------------|----------------|
| Baseline | 0.951 | 0.535 | 0.979 | 0.986 | **0.595** |
| Ours | **0.954** | **0.541** | **0.983** | **0.989** | 0.557 |

Our approach demonstrates strong generalization capabilities beyond its training domain of static scenes with camera motion, effectively improving performance on diverse benchmarks with highly dynamic scenes.

Evaluation on VBench 2.0, MiraData and VideoReward benchmarks using challenging general prompts shows consistent improvements across all metrics. Our method achieves 57.9% overall win rate on VBench 2.0 and 58.5% on VideoReward, with particularly strong performance in visual quality (61.3% and 65.0% respectively) and motion quality (55.3% and 58.5% respectively). Remarkably, our model maintains similar performance (58.5% overall) on MiraData videos with dynamic objects, demonstrating robust generalization across general benchmarks despite training only on static scenes.

This generalization occurs because aligning models with smoother, geometrically consistent camera trajectories inherently improves video quality even when objects move independently. The primary sources of error in dynamic scenes: unstable motion trajectories, artifacts, and flickering are amplified by object movement. By learning to improve 3D consistency, our trainable adapter addresses these issues, thereby automatically enhancing the quality of dynamic object generation. VBench metrics confirm that geometric consistency training benefits transfer effectively across diverse scenarios, with improvements in background consistency, temporal stability, and motion smoothness validating our core insight that classical geometric constraints enhance overall 3D understanding.

## 4.5 ABLATION STUDY

We ablate descriptor choices, geometric metrics, alignment methods, and design components. For efficiency the ablations are done on a subset of data.

Table 4: **Metric Ablation:** Simple descriptors with classical metrics achieve balanced performance, while sophisticated descriptors can be counterproductive for video alignment.

| Descriptor | Metric | Visual Quality | Motion Quality | Text Alignment | Overall |
|------------|--------|----------------|----------------|----------------|---------|
| SIFT | Sampson Error | 64.3% | 64.2% | 41.8% | 57.1% |
| LightGlue | Sampson Error | 70.3% | 52.6% | 38.5% | 53.8% |
| SEA-Raft | Sampson Error | 80.3% | 56.0% | 33.6% | 56.9% |
| SIFT | Symmetric Epipolar | 76.4% | 59.6% | 36.4% | 56.4% |

**Descriptor and Metric Analysis:** While SEA-Raft achieves highest visual quality (80.3%), we observe it hacks the reward by preferring oversaturated scenes. LightGlue finds good correspondences in clean areas when videos contain artifacts, resulting in misleadingly low epipolar error, whereas we want correspondences across the entire scene so artifacts anywhere produce high error. Generally,

Table 5: Win-rate on the VideoReward benchmark comparing different finetuning strategies with geometric consistency metrics.

| Method | VideoReward Metrics | | | | Consistency Metrics | | |
|---|---|---|---|---|---|---|---|
| | VQ | MQ | TA | Overall | Perspective ↑ | Sampson ↓ | Dynamics ↓ |
| SFT | 66.0% | 63.0% | 54.0% | 64.5% | 0.427 | 0.161 | 0.225 |
| Flow-RWR (Liu et al., 2025) | 63.5% | 60.5% | **57.0%** | 64.0% | **0.434** | 0.174 | 0.229 |
| DRO (Li et al., 2025b) | 65.0% | 54.0% | 50.5% | 64.5% | 0.410 | **0.068** | **0.195** |
| Epipolar-DPO (Ours) | **72.0%** | **71.0%** | 55.0% | **73.0%** | 0.428 | 0.127 | 0.223 |

all setups are comparable, but our main claim is that classical geometric constraints provide cleaner optimization signals than sophisticated alternatives that can miss global inconsistencies.

**Comparison with Learnable Metrics:** Models trained with VideoReward Motion Quality achieve 61.3% VideoReward but 0.179 Sampson error, while ours achieve 64.3% VideoReward and 0.131 Sampson error, superior on both metrics. Similarly, MET3R-trained models achieve 0.049 MET3R score but 0.176 Sampson error, while Sampson-trained models match this MET3R score (0.049) with far better Sampson performance (0.131). This confirms learnable metrics produce insufficient preference signals compared to classical geometry constraints.

**Static Penalty Analysis:** The temporal variation penalty achieves superior motion dynamics (dynamic degree 0.710 vs 0.627) while maintaining comparable geometric consistency and motion quality. This component effectively prevents degenerate static solutions while preserving the geometric alignment. We do observe that large $\lambda$ in temporal variation cause the model to always increase the camera motion; hurting both 3D consistency and generalization. Similarly, decreasing the lambda forces the model to optimize for a naive solution of increasing 3D consistency by reducing the motion.

Table 6: **Static Penalty Ablation:** Adding static penalty significantly improves dynamic degree while only slightly sacrificing the consistency.

| Method | Dynamic Degree | Motion Quality | Sampson Error |
|---|---|---|---|
| Ours | 0.627 | **71.5%** | **0.127** |
| Ours + Static Penalty | **0.710** | 69.5% | 0.131 |

**Alignment Method Comparison:** Our approach outperforms all alternatives with highest win rates on VideoReward, validating effectiveness of DPO for video model optimization. We compare the alignment against two simple methods: Supervised Finetuning on Win samples, and Flow-RWR (Liu et al., 2025) which additionally weights the loss with the normalized reward. All strategies are trained with the same number of steps and target metric. We also find DRO (Li et al., 2025b) interesting as it does not require regularization with the reference model, but in our case, it quickly leads to degenerate solutions deviating from the base model and producing strong artifacts. The other two setups, while still effective, can not learn from the gap between consistent and inconsistent trajectories, resulting in less 3D consistent outputs. The evaluation also showcases the necessity of evaluating the model with multiple metrics. For example, learnable metrics are subject to overfitting: a model with strong artifacts still can get high scores from a VLM-based metric.

## 5 CONCLUSION

We present a novel approach for enhancing 3D consistency in video diffusion models by leveraging epipolar geometry constraints as preference signals. Our work shows that classical geometric constraints provide more stable optimization signals than modern learned metrics, which produce noisy targets that compromise alignment quality, and training on static scenes generalizes effectively to diverse dynamic content, demonstrating the broad applicability of geometric principles. The resulting models generate videos with fewer geometric inconsistencies and more stable camera trajectories while preserving creative flexibility. This work highlights how classical computer vision algorithms effectively complement deep learning approaches, addressing limitations in purely data-driven methods and improving content quality through adherence to fundamental physical principles.

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

## A Visual Quality Evaluation

Table 7: Visual Quality and Aesthetic Fidelity Results (Text-to-Video)

| Method | VBench Visual Metrics | | VideoReward | Human Eval |
|---|---|---|---|---|
| | Background Consistency ↑ | Aesthetic Quality ↑ | Visual Quality ↑ | Visual Preference Rate |
| Baseline | 0.930 | 0.541 | - | 15.0% |
| Ours | **0.942** | **0.551** | **72.0%** | **52.8%** |

Table 8: **Win-rate vs. Wan-2.1-14B (Wang et al., 2025a)** on the VideoReward (Liu et al., 2025) benchmark. The Baseline and Epipolar-Aligned Model contain only 1.3B parameters.

| | Text-to-Video | | | |
|---|---|---|---|---|
| Method | Visual Quality | Motion Quality | Text Alignment | Overall |
| Baseline | 13.3% | 14.4% | 24.2% | 8.6% |
| DPO-Epipolar | **18.1%** | **21.8%** | **25.0%** | **13.8%** |

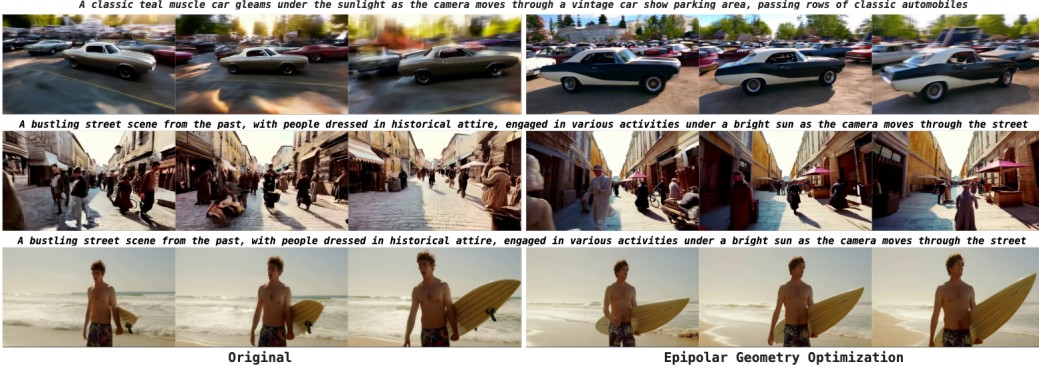

Figure 4: **Qualitative Evaluation:** Comparison of baseline and epipolar-aligned models on dynamic scenes featuring both camera movement and object motion. Our approach maintains improved geometric consistency and smoother trajectories, demonstrating generalization beyond static scene training. Best seen in the supplementary video.

Generating more geometrically consistent scenes with fewer artifacts naturally leads to higher overall visual quality of the generated content. To validate this connection, we evaluate visual fidelity across multiple assessment frameworks.

**Aesthetic Metrics:** VBench (Huang et al., 2024) visual quality assessment shows consistent improvements across multiple dimensions. For example, background consistency increases from 0.930 to 0.942 and aesthetic quality improves from 0.541 to 0.551. These metrics confirm that geometric training enhances visual stability and perceived quality.

**Perceptual Assessment:** VideoReward (Liu et al., 2025) visual quality evaluation demonstrates substantial improvement with a 72.0% win rate, indicating that human-distilled quality assessment strongly favors our geometrically-aligned approach. This suggests that geometric consistency contributes significantly to overall visual appeal.

**Human Validation:** Human preference evaluation shows a 52.8% preference rate for our method's visual quality across all video types, further validating that geometric improvements translate to perceptually superior results that human evaluators can identify and prefer.

## B SUPERVISED FINETUNING EVALUATION

We compare the effectiveness of epipolar-aware alignment with standard video model finetuning on multi-view static datasets (Ling et al., 2024; Zhou et al., 2018). Directly finetuning the model on perfectly 3D-consistent videos of static scenes could be seen as an alternative approach to improve generation consistency. Yet this naive approach only trains the model to reproduce observed frames, thereby optimizing average risk. It is not directly optimizing task-level objectives, such as multi-view geometric consistency, and instead might replicate biases present in the dataset and regress toward "mean" solutions for ambiguous problems. We observe that models finetuned with this objective overfit to dataset-specific factors, such as a narrow set of camera trajectories in RealEstate-10K and DL3DV videos and specific scene types, which hurts the generalization capabilities of the base model. Table 9 demonstrates that a model directly finetuned on multi-view datasets does not generalize well to dynamic scenes and, on average, tends to amplify camera motion, mimicking the average training set trajectories.

Table 9: **Comparison vs. Supervised Finetuning on Multi-View Data.** Finetuning on real multi-view static scenes (Zhou et al., 2018; Ling et al., 2024) propagates dataset bias such as scene type or specific camera motion. As a result the model overfits to dataset specifics rather than learning general principles.

| Benchmark | Method | Visual Quality | Motion Quality | Text Alignment | Overall |
|---|---|---|---|---|---|
| VideoReward | Supervised Finetuning | 37.25% | 46.75% | **53.0%** | 44.75% |
| | DPO-Epipolar | **65.0%** | **58.5%** | 50.5% | **58.5%** |
| VBench | Supervised Finetuning | 35.8% | 53.0% | 39.4% | 35.2% |
| | DPO-Epipolar | **61.3%** | **55.3%** | **52.0%** | **57.9%** |
| MiraData9K | Supervised Finetuning | 41.5% | 42.0% | 51.0% | 47.0% |
| | DPO-Epipolar | **57.0%** | **58.0%** | **52.0%** | **58.5%** |

## C IMAGE-TO-VIDEO EVALUATION

Table 10: **Image-to-Video Alignment:** Despite image conditioning constraints, epipolar alignment shows consistent improvements across multiple metrics.

| Method | VideoReward | | 3D Reconstruction | | | 3D Consistency | | VBench Metrics | | | | |
|---|---|---|---|---|---|---|---|---|---|---|---|---|
| | Visual Quality | Motion Quality | PSNR ↑ | SSIM ↑ | LPIPS ↓ | Motion (SSIM ↓) | Sampson Error ↓ | Background Consistency | Aesthetic Quality | Temporal Flickering | Motion Smoothness | Dynamic Degree |
| Baseline | - | - | **21.08** | 0.686 | 0.408 | 0.239 | 0.215 | 0.955 | 0.498 | **0.981** | 0.992 | **0.378** |
| Ours | **51.35%** | **56.08%** | 20.99 | **0.700** | **0.377** | **0.239** | **0.197** | **0.955** | **0.499** | 0.980 | **0.992** | 0.343 |

Image-to-video alignment presents unique challenges due to the strong conditioning signal from the input image. The image conditioning is integrated into intermediate layers of the diffusion process, creating additional constraints that naturally reduce output variance and make alignment more challenging. Despite these limitations, our epipolar geometry optimization demonstrates consistent positive impact across multiple evaluation dimensions.

The 3D reconstruction results validate the geometric improvements: SSIM improves from 0.686 to 0.700, and LPIPS decreases from 0.408 to 0.377. These gains, while more modest than text-to-video results, confirm that enhanced geometric consistency translates to better downstream 3D understanding even under image conditioning constraints. The Sampson epipolar error improvement from 0.215 to 0.197 further validates the effectiveness of classical geometric alignment.

VideoReward metrics show meaningful improvements in motion quality (56.08% vs 43.92%) and visual quality (51.35% vs 48.65%). VBench metrics remain stable with slight improvements in aesthetic quality, demonstrating that geometric optimization preserves overall generation quality while enhancing 3D consistency.

While the input image provides strong structural guidance, it also constrains the model's ability to adapt toward geometrically optimal solutions. Nevertheless, consistent positive trends across

reconstruction, consistency, and quality metrics validate that classical geometric constraints provide reliable optimization signals even in constrained generation scenarios.

## D    PROMPT OPTIMIZATION EVALUATION

We further compare our method with VPO (Cheng et al., 2025), a video prompt optimization technique, which is complementary to our approach since it optimizes prompts rather than model weights. We evaluate VPO alone and in combination with our method. Results are reported in Table 11.

Table 11: **Prompt Optimization:** VPO optimizes prompts while our method improves geometry alignment. The two approaches are complementary and can be combined to achieve both high visual quality and geometric consistency.

| Method | Visual Quality ↑ | Motion Quality ↑ | Overall ↑ | Dynamic Degree ↑ | Motion (mean SSIM) ↓ |
|---|---|---|---|---|---|
| Ours | 63.1% | 65.8% | 59.1% | **0.80** | **0.211** |
| VPO | 59.1% | 70.6% | 82.7% | 0.65 | 0.235 |
| VPO + Ours | **67.0%** | **71.9%** | **83.6%** | 0.61 | 0.234 |

We observe that VPO tends to reduce camera motion and restructure prompts while optimizing for general video quality. However, such prompt optimization methods can be efficiently combined with geometry-aligned models like ours to simultaneously achieve high visual quality and geometric consistency.

## E    SCALING ANALYSIS

To understand how our geometric alignment performs across different model scales, we compare both the baseline and epipolar-aligned 1.3B parameter models against the much larger Wan-2.1-14B model (Wang et al., 2025a). As shown in Table 8, while the performance gap remains substantial due to the 14B model's higher resolution (720p) and superior base capabilities, our epipolar alignment helps close this gap meaningfully. The aligned 1.3B model achieves win rates of 18.1%, 21.8%, and 25.0% for Visual Quality, Motion Quality, and Text Alignment respectively, compared to 13.3%, 14.4%, and 24.2% for the baseline 1.3B model. Notably, the 14B model requires approximately 10× longer inference time than the 1.3B variant, making our alignment approach particularly valuable for applications where computational efficiency is critical. This suggests that geometric consistency improvements can partially compensate for scale limitations, offering a practical path toward better video quality without the computational overhead of significantly larger models.

## F    QUALITATIVE EVALUATION

For comprehensive assessment of video quality and geometric consistency, we include an interactive webpage in the supplementary materials where readers can view the full video sequences and directly compare the baseline and epipolar-aligned model outputs.

## G    THE USE OF LARGE LANGUAGE MODELS (LLMS)

The LLM is used as a core part of the prompt expansion method 3 ensuring we build a large library of diverse captions with challenging camera trajectories for preference data generation. We also used LLMs to polish the writing, verify grammar or improve the sentence structure.

## H    LIMITATIONS AND BROADER IMPACT

Our approach primarily focuses on static scenes with dynamic camera movements, aligning well with applications in 3D reconstruction and novel view synthesis. Adapting this method to scenes with dynamic objects would require modifying the training pipeline to separately model and evaluate

object motion and camera movement. Additionally, epipolar geometry constraints assume point correspondences coming from a static scene under camera motion, limiting effectiveness for scenes with independent object movement or non-rigid deformations where a single fundamental matrix cannot explain all correspondences. The data mining relies on feature matching and epipolar geometry scoring. However, it can produce false positives (assigning low error to geometrically inconsistent videos) and false negatives (high error to consistent ones) when scenes have repetitive textures, lack distinctive features, or contain significant motion blur. In addition, while aigned model significantly improves geometric consistency, it does not solve all failure modes. Complex dynamic scenes with extreme camera motion or highly ambiguous content remain challenging for both the baseline and our approach. Video generation models may be misused to produce realistic but deceptive content, contributing to the spread of misinformation, political manipulation, and erosion of public trust. Furthermore, the computational resources required to train such models raise environmental concerns and may exacerbate inequalities in access to advanced AI technologies. Geometry-aware video generation can facilitate various 3D vision tasks, including scene reconstruction, SLAM, and visual odometry. By improving geometric consistency in generated videos, our method produces more realistic and usable synthetic data for training computer vision systems. This advances applications in robotics and autonomous navigation, where accurate spatial understanding is crucial. The integration of classical geometry principles with modern generative models represents a promising direction for enhancing AI systems with stronger physical world understanding.

