# OpenReview forum: "Epipolar Geometry Improves Video Generation Models"
_ICLR.cc/2026/Conference — Submitted to ICLR 2026_

### Official Review · Reviewer_dEzy · 2025-10-29

**Soundness:** 3
**Presentation:** 3
**Contribution:** 3
**Rating:** 2
**Confidence:** 4

**Summary:**

The paper proposes improving video diffusion models by introducing epipolar geometry constraints through preference-based optimization. Although large video generation models trained with rectified flow show impressive visual quality, they often suffer from geometric inconsistency, unstable motion, and artifacts. The method enforces classical geometric principles to stabilize camera motion and reduce distortions—without requiring differentiable rendering.

**Strengths:**

1. The problem addressed is highly important — achieving geometrically consistent video generation remains a major challenge today.

2. The proposed method is simple yet effective.

3. The generated results show clear improvements compared to the baseline.

4. The project website is beautifully designed. If possible, could you let me know which GitHub template it is based on?

**Weaknesses:**

1.	How should we evaluate 3D consistency when videos inevitably contain dynamic content? If the presence of motion leads to degraded 3D consistency and such data are consequently filtered out, what would be the impact of suppressing dynamic content generation?

2.	Currently, the method is only compared against a single baseline (the original video generation model). However, I think an important baseline is missing — namely, fine-tuning Wan 2.1 with DL3DV and Real10K using LoRA. My reasoning is that works like DimensionX[1] have already demonstrated that, for tasks such as novel view synthesis from a single image, fine-tuning video diffusion models on such datasets alone can yield geometrically consistent generations. Therefore, in this work, it is difficult to determine whether the improvement in core capability truly comes from the proposed epipolar geometry optimization, or simply because the training and test datasets share the same distribution. I also believe that a visual comparison with this baseline would be important. This is my main concern. If fix this problem, I will raise my score.
[1] DimensionX: Create Any 3D and 4D Scenes from a Single Image with Controllable Video Diffusion

3. My understanding of the data fine-tuning pipeline is as follows: for each prompt, three videos are sampled, their scores are computed, and two of them are selected for DPO fine-tuning. Please correct me if I am mistaken. How long does the scoring process take — is it fast? Why are only three videos sampled instead of more? Why do videos with the same caption show noticeable differences in 3D quality? Can the DPO process use GT videos as the higher-score samples? If GT videos are used as high-score references, how does that differ from directly fine-tuning with GT data?

4. Regarding evaluation metrics, the baseline videos clearly show  geometric distortion, and artifacts. Given the obvious visual differences, why do the metrics in Table 1 — apart from Human Eval — show such limited improvement? Does this imply that the chosen metrics are not appropriate, that the evaluation method has limitations, or that even the optimized results still fail to achieve satisfactory 3D reconstruction quality? How large is the quantitative gap compared to using GT videos directly for reconstruction?

5. Could you elaborate on the setting of Table 5 — for example, what rewards were used and which strategy led to the observed performance differences? I find this comparison particularly interesting.

6. Regarding the proposed large-scale preference dataset, what do you think is its actual value? Since all the data are generated by video models, and given that Wan 2.1 (1.3B) still shows a significant performance gap from real data, why do we need these imperfect datasets?

**Questions:**

Please refer to the weeknesses.

---

> ### Author Response · Authors · 2025-11-24
> **Official Comment by Authors**
>
> Thank you for your review. We appreciate your concerns, have conducted additional experiments to address them and provide detailed responses below.
>
> > **The project website is beautifully designed. If possible, could you let me know which GitHub template it is based on?**
>
> Thank you! The webpage is inspired by the [Segment Anything website](https://aidemos.meta.com/segment-anything), we will release the website code together with the code and models to let others reuse the template.
>
> > **How should we evaluate 3D consistency when videos inevitably contain dynamic content? If the presence of motion leads to degraded 3D consistency and such data are consequently filtered out, what would be the impact of suppressing dynamic content generation?**
>
> ***Training:*** We train only using static scenes.  However, one of the main insights of our work is that aligning the model on a large set of static scenes transfers to dynamic videos!
> Dynamic generated videos also improve considerably in 3D consistency.
> Our approach learns from a 3D consistency gap with implicit KL-divergence regularization, forcing the model to focus on optimizing the camera motion and reducing the artifacts while preserving strong generalization to other types of scenes.
>
> ***Evaluation:*** This is also a reason why we split the evaluation. Section 4.2 and Table 1 evaluated the model on static scenes, eliminating the impact of dynamic content. Section 4.4 and Table 3 evaluate generalization to dynamic content on standard benchmarks (VBench, VideoReward) that compute a wide variety of motion and content quality.
>
> > **Currently, the method is only compared against a single baseline (the original video generation model). However, I think an important baseline is missing — namely, fine-tuning Wan 2.1 with DL3DV and Real10K using LoRA. My reasoning is that works like DimensionX[1] have already demonstrated that, for tasks such as novel view synthesis from a single image, fine-tuning video diffusion models on such datasets alone can yield geometrically consistent generations. [....] I also believe that a visual comparison with this baseline would be important.**
>
> Thanks for the question! We have conducted this experiment for the rebuttal: we fine-tune Wan 2.1 with these datasets and observe that the aligned model quickly overfits, consistently generating the same forward-facing trajectory regardless of the scene type, which limits the model's generalization.
> This is the key difference between supervised finetuning and RL / DPO alignment.
> In contrast, DPO alignment enforces an implicit KL-divergence constraint, which preserves the generalization and the self-training routine on expanded prompts of various motions, ensuring we learn in highly varied scenes encoding more general principles.
> We have added this experiment and an extended discussion to Appendix B. In addition, we do provide the visual comparison in the updated anonymous webpage: https://anon-epipolar-dpo.github.io/anon-epipolar-dpo/
>
> > **Therefore, in this work, it is difficult to determine whether the improvement in core capability truly comes from the proposed epipolar geometry optimization, or simply because the training and test datasets share the same distribution.**
>
> The reason we evaluate the model both on static scenes from these datasets as well as generic dynamic benchmarks is to ensure that the improvement doesn't come just from the similarity between the train and test sets. We observe the opposite -- the model directly finetuned on multi-view datasets tends to replicate the biases present in the datasets.
>
> > **How long does the scoring process take — is it fast?**
>
> Yes, compared to video generation, scoring is near instant (less than 500ms).
>
> > **Why are only three videos sampled instead of more?**
>
> We observe that for some scenes, generating more samples does not result in a significant difference between predictions, so we favor generating more scenes and filtering out those that do not pass the quality threshold rather than generating fewer scenes with more samples.

---

> > ### Author Response · Authors · 2025-11-24
> > **Official Comment by Authors**
> >
> > > **Why do videos with the same caption show noticeable differences in 3D quality?**
> >
> > This is the exact property of diffusion models that makes alignment effective! Due to the stochastic nature of the diffusion sampling process, different noise initializations lead to different denoising trajectories through the learned latent space. Even with identical text conditioning, this results in samples from different regions of the distribution, where some trajectories produce outputs with better geometric consistency and 3D coherence than others. In a sense, the model has learned a too wide distribution, which we can sharpen with our objective.
> >
> > > **Can the DPO process use GT videos as the higher-score samples? If GT videos are used as high-score references, how does that differ from directly fine-tuning with GT data?**
> >
> > While it is possible to use real data through noise inversion, DPO assumes both samples in a pair come from the model's distribution (or a reference model's distribution). GT videos come from a fundamentally different distribution (real data) that the diffusion model may assign a very low probability to. The optimization assumes you can trade off between the samples - but if GT is far outside what the model can generate, it cannot learn a meaningful preference direction.
> >
> > > **Regarding evaluation metrics, the baseline videos clearly show geometric distortion, and artifacts. Given the obvious visual differences, why do the metrics in Table 1 — apart from Human Eval — show such limited improvement? Does this imply that the chosen metrics are not appropriate, that the evaluation method has limitations, or that even the optimized results still fail to achieve satisfactory 3D reconstruction quality? How large is the quantitative gap compared to using GT videos directly for reconstruction?**
> >
> > As in every dataset, a large part of the test samples are "easy samples" for which the base model already produces stable outputs. This is also why we perform a human evaluation, which focuses on hard samples. In our opinion, the public benchmarks contain too many easy examples, which dilute the performance gap between methods.
> >
> > > **Could you elaborate on the setting of Table 5 — for example, what rewards were used and which strategy led to the observed performance differences? I find this comparison particularly interesting.**
> >
> > We test 2 simple strategies: Supervised Finetuning on Win samples, and Flow-RWR which additionally weights the loss with the normalized reward. All strategies are trained in the same setup: Epipolar error as the metric, the same number of steps, etc. We also find DRO interesting as it does not require regularization with the reference model (significantly improving the training time), but in our case, it quickly leads to degenerate solutions deviating from the base model and producing strong artifacts.
> > The other two setups, while still effective, can not learn from the gap between consistent and inconsistent trajectories, resulting in less 3D consistent outputs. The evaluation also showcases the necessity of evaluating the model with multiple metrics. For example, learnable metrics are subject to overfitting: a model with strong artifacts still can get high scores from a VLM-based metric. We have expanded this discussion in the paper.
> >
> > > **Regarding the proposed large-scale preference dataset, what do you think is its actual value? Since all the data are generated by video models, and given that Wan 2.1 (1.3B) still shows a significant performance gap from real data, why do we need these imperfect datasets?**
> >
> > This is a great question. By far the most computationally expensive step of our paper is the data generation, which dwarves model training. Releasing the generated videos will allow others to experiment with more complex or learnable 3D consistency metrics or training setups without the need for generating more videos.
> > More importantly, we believe that the general pipeline, including a large library of captions with amplified camera motion, will allow us to expand the research to other video models or tasks. We have clarified this in the revised manuscript.

---

> > > ### Author Response · Authors · 2025-11-24
> > > **Comparison with supervised finetuning**
> > >
> > > | Benchmark | Method | Visual Quality | Motion Quality | Text Alignment | Overall |
> > > |-----------|--------|----------------|----------------|----------------|---------|
> > > | VideoReward | Supervised Finetuning | 37.25% | 46.75% | **53.0%** | 44.75% |
> > > | VideoReward | DPO-Epipolar | **65.0%** | **58.5%** | 50.5% | **58.5%** |
> > > | VBench | Supervised Finetuning | 35.8% | 53.0% | 39.4% | 35.2% |
> > > | VBench | DPO-Epipolar | **61.3%** | **55.3%** | **52.0%** | **57.9%** |
> > > | MiraData9K | Supervised Finetuning | 41.5% | 42.0% | 51.0% | 47.0% |
> > > | MiraData9K | DPO-Epipolar | **57.0%** | **58.0%** | **52.0%** | **58.5%** |

---

### Official Review · Reviewer_67U8 · 2025-10-31

**Soundness:** 3
**Presentation:** 2
**Contribution:** 2
**Rating:** 6
**Confidence:** 4

**Summary:**

The paper proposes to align pretrained video diffusion models with epipolar geometry via preference-based finetuning to enforce the 3D consistency of the generated video. More specifically, it samples multiple videos per prompt, ranks them using a Sampson epipolar error computed from SIFT/RANSAC correspondences, and applies Flow-DPO with a small temporal-variation penalty to avoid degenerated solutions such as frozen scenes. The experiment results shows that the epipolar geometry based preference alignment leads to better 3D consistency of the generated video.

**Strengths:**

+The idea of rank-based DPO alignment with classical geometry is interesting. The paper applies Direct Preference Optimization (Flow-DPO) so the model learns from pairwise rankings induced by Sampson epipolar error—no absolute, differentiable reward needed. This leverages DPO’s pairwise nature (relative preferences per prompt) and enables efficient latent-space LoRA finetuning without decoding full videos, which is practically useful for video models.

+LoRA-based Flow-DPO with an explicit static-penalty term (Eq. 3) is easy to reproduce and mitigates the “frozen scene” failure mode.

**Weaknesses:**

-Dynamic objects are not handled. The approach assumes static scenes for the reward; moving objects violate a single fundamental matrix and can corrupt the signal.This is important because many real prompts include independent object motion.

-Dynamics–consistency trade-off. Despite the temporal-variation penalty, dynamic degree drops (Table 2), suggesting a residual tendency to damp motion. The paper frames this as an acceptable trade-off but doesn’t systemically tune λ or add complementary rewards to preserve motion amplitude.

**Questions:**

1. Can the methods be applied on training videos with dynamic objects? For those videos, as long as we can extract the camera poses, and segment the dynamic objects to rule out those objects during reward calculation, the proposed method could be applied. This will greatly increase the breadth of the application.

2. Building the offline preference set (≈54k videos; ~1,980 GPU-hrs) is non-trivial; discussion of scalability, deduping, and potential bias in prompt expansions would help practitioners.

---

> ### Author Response · Authors · 2025-11-24
> **Official Comment by Authors**
>
> Thank you for the thoughtful and positive review of our work! Below we address the main questions and concerns:
>
> > **Dynamic objects are not handled. The approach assumes static scenes for the reward; moving objects violate a single fundamental matrix and can corrupt the signal.This is important because many real prompts include independent object motion.**
>
> Generalization to dynamic scenes is exactly one of the main insights of our work. Aligning the model on a large set of static scenes ensures the robustness of the metric, while learning from a 3D consistency gap with implicit KL-divergence regularization forces the model to focus on optimizing the camera motion and reducing the artifacts while preserving the strong generalization to other types of scenes. Thus, a model trained in this setup can generalize even to dynamic videos. Additionally, we hypothesize that the LoRA adapter (2.5% of total model weights) is small enough to learn a generalizable modification/shaping of the model's sample distribution that is independent of the video content.
>
> > **Dynamics–consistency trade-off. Despite the temporal-variation penalty, dynamic degree drops (Table 2), suggesting a residual tendency to damp motion. The paper frames this as an acceptable trade-off but doesn’t systemically tune $\lambda$ or add complementary rewards to preserve motion amplitude.**
>
> The dynamics–consistency is one of the core parts of our analysis (Table 2). We believe that in many cases, reducing motion is crucial for the 3D consistency (see webpage videos: Basketball Court, Car Show). We also observe that tuning λ results in either solution that always reduces or amplifies the motion, hurting either quality or generalization. We have added this insight to the revised manuscript.
>
> > **Can the methods be applied on training videos with dynamic objects? For those videos, as long as we can extract the camera poses, and segment the dynamic objects to rule out those objects during reward calculation, the proposed method could be applied. This will greatly increase the breadth of the application.**
>
> This is an intriguing suggestion. However, it is somewhat of a chicken and egg problem: to estimate camera poses and to detect dynamic objects, we would need to be able to run some form of SfM method on the data. These methods typically struggle with dynamic sequences, so the realization of this idea might be complicated. Camera pose estimation for dynamic sequences remains an open problem; however, as soon as these methods become stable, they will be fully compatible with our method. Yet, our method demonstrates that aligning the model on a large set of static scenes transfers to dynamic videos.
>
> > **Building the offline preference set (≈54k videos; ~1,980 GPU-hrs) is non-trivial; discussion of scalability, deduping, and potential bias in prompt expansions would help practitioners.**
>
> We agree. This is also one of the core steps of the pipeline, which ensures we can learn from a meaningful gap. This is why we dedicated a lot of attention to parts such as prompt expansion (Section 3.3). We will release the code and trained model for full reproducibility.

---

### Official Review · Reviewer_zotX · 2025-10-31

**Soundness:** 3
**Presentation:** 3
**Contribution:** 2
**Rating:** 4
**Confidence:** 4

**Summary:**

The paper introduces a method for enhancing the 3D consistency of video generation models by using pairwise epipolar geometry constraints and preference-based optimization without requiring end-to-end differentiability, and reducing unstable camera trajectories and geometric artifacts through mathematically principled geometric enforcement. The paper demonstrates that using classical geometric constraints, epipolar geometry, provides simple but stable optimization signals, leading to improved video generation quality and enhanced 3D consistency. The method is evaluated across several metrics and performs well, including 3D consistency, motion stability, and generalization across dynamic scenes.

**Strengths:**

- The incorporation of classical epipolar geometry into modern video generation models to improve 3D consistency is interesting.
- The paper presents a thorough evaluation framework that assesses various aspects of video generation quality, which provides a holistic view of the model’s performance and demonstrates the practical benefits of the proposed method. The results show that the use of epipolar geometry reduces geometric artifacts and improves motion smoothness.
- The method also generalizes effectively to dynamic content, despite being trained primarily on static scenes with dynamic cameras.

**Weaknesses:**

- The performance of the proposed method is limited by the capabilities of the base model itself. The training data is generated by the base model and then sorted. Therefore, the upper bounds of 3D consistency and motion complexity are relatively fixed, even if the input prompts are more complex. Due to the inherent randomness of the base model, generating three videos per caption, I would question the proportion of high-quality samples. I am also curious why this paper does not use labeled real videos as the ground truth to further improve 3D consistency.
- Epipolar geometry is typically only applicable to static scenes. It will fail to work with dynamic objects or non-rigid bodies, which are common in real-world scenarios, thus causing problems when scaling up the data.

**Questions:**

- What is the impact of the hyperparameter $\lambda$ in the temporal variation loss on the final video quality and 3D consistency? A more detailed analysis would be beneficial to understand the trade-offs between motion stability and 3D consistency.
- Theoretically, epipolar geometry constraints cannot effectively constrain dynamic objects, but the results show that even models trained only on static scenes can generalize to dynamic scenes. I'm curious about the deeper insights into how the model acquires this ability.
- I noticed that most of the dynamic objects in the demo are rigid bodies with relatively small movements. What if we were dealing with more extreme but common real-world scenarios? For example, large motions, non-rigid deformations, or sudden occlusions.

---

> ### Author Response · Authors · 2025-11-24
> **Official Comment by Authors**
>
> Thank you for your review. We appreciate your feedback and provide detailed responses to the main concerns and questions below.
>
> > **The performance of the proposed method is limited by the capabilities of the base model itself. The training data is generated by the base model and then sorted. Therefore, the upper bounds of 3D consistency and motion complexity are relatively fixed, even if the input prompts are more complex.**
>
> This is a fundamental property of all approaches of this type. However, the underlying assumption is that the original model has learned a broader distribution than the data needs. Thus, reranking and fine-tuning sharpen the distribution, thereby improving the overall average sample quality by increasing the likelihood of good samples and decreasing the likelihood of bad samples. Our experiments show that this indeed consistently improves the model.
> Moreover, adding new external training data often biases the model towards that data distribution, whereas generated samples are inherently from within the model's distribution.
>
> > **Due to the inherent randomness of the base model, generating three videos per caption, I would question the proportion of high-quality samples. I am also curious why this paper does not use labeled real videos as the ground truth to further improve 3D consistency.**
>
> The proportion of high-quality training triplets is 70%. This is precisely why we include extensive filtering to ensure we only extract pairs with a meaningful 3D consistency gap.
> As mentioned above, real data typically introduces bias where the model learns to fit the new data distribution of the real data at the same time as learning the objective. Using generated samples is less prone to this problem as all samples come from within the model's distribution. For the rebuttal, we have evaluated the performance of the model finetuned on real videos from DL3DV and RealEstate-10K and did find that the model tends to overfit to dataset specifics. The results are added to Table 9 and Supplementary B and further prove the effectiveness of our approach.
>
> > **Epipolar geometry is typically only applicable to static scenes. It will fail to work with dynamic objects or non-rigid bodies, which are common in real-world scenarios, thus causing problems when scaling up the data.**
>
> This is correct, and the reason we train only using static scenes.
> However, one of the main insights of our work is that aligning the model on a large set of static scenes transfers to dynamic videos!
> Dynamic generated videos also improve considerably in 3D consistency.
> Our approach learns from a 3D consistency gap with implicit KL-divergence regularization, forcing the model to focus on optimizing the camera motion and reducing the artifacts while preserving strong generalization to other types of scenes.
>
> > **What is the impact of the hyperparameter in the temporal variation loss on the final video quality and 3D consistency? A more detailed analysis would be beneficial to understand the trade-offs between motion stability and 3D consistency.**
>
> Thanks for the question. We observe that a large $\lambda$ in temporal variation cause the model to always increase the camera motion; hurting both 3D consistency and generalization. Similarly, training without the loss or with very low **lambda** forces the model to optimize for a naive solution of increasing 3D consistency by reducing the motion. We have added this analysis to the revised manuscript.
>
> > **Theoretically, epipolar geometry constraints cannot effectively constrain dynamic objects, but the results show that even models trained only on static scenes can generalize to dynamic scenes. I'm curious about the deeper insights into how the model acquires this ability.**
>
> We train only using static scenes. We hypothesize that the LoRA adapter (2.5% of total model weights) is small enough to learn a generalizable modification/shaping of the model's sample distribution that is independent of the video content. This could be an indication that the original base model has learned a disentangled representation for content and consistency, allowing the adapter to generalize. A full mechanistic interpretation of the original and adapted model would be highly interesting, but goes beyond the scope of this paper.
>
> > **I noticed that most of the dynamic objects in the demo are rigid bodies with relatively small movements. What if we were dealing with more extreme but common real-world scenarios? For example, large motions, non-rigid deformations, or sudden occlusions.**
>
> Our alignment strategy focuses on reducing the artifacts and improving camera motion in the base model. Yet, the very complex scenes with large motions can simply be out-of-distribution for the base video model. We have added the analysis of the common failure cases to the updated anonymized webpage: https://anon-epipolar-dpo.github.io/anon-epipolar-dpo/

---

### Official Review · Reviewer_QaX5 · 2025-11-01

**Soundness:** 3
**Presentation:** 3
**Contribution:** 2
**Rating:** 4
**Confidence:** 4

**Summary:**

This paper improves video diffusion models by incorporating epipolar geometry into the finetuning process. Using Sampson distance to rank candidate generations, the method performs DPO-based preference alignment to encourage 3D-consistent video generation. The authors evaluate extensively across geometric accuracy, video quality, reconstruction, and human preference, showing meaningful improvements in temporal and geometric stability.

**Strengths:**

- Combining classical geometry with generative models is a good idea and helps pushing video diffusion models toward physically grounded behavior.
- Extensive evaluation protocol makes the empirical results convincing.
- The paper is well structured, clear, and easy to follow. Also it clearly represents limitations and broader impact discussion in Appendix F.
- Preparing a large-scale geometry preference dataset is a significant effort, and publicly releasing it will be useful to the community.

**Weaknesses:**

- Dynamic degree decreases, showing that the method improves stability partly by reducing motion amplitude. There is a trade-off between temporal consistency and keeping lively, dynamic motion.
- Although better than baseline, some videos still show flickering, blur, and occasional scene drift, especially in challenging or texture-heavy regions.
- The dataset creation pipeline is computationally heavy and slow (as acknowledged in Appendix F), making the method difficult to scale or replicate.

**Questions:**

- Instead of relying only on pairwise epipolar distance, could multi-frame geometric reasoning (e.g., triangulation error, bundle-adjustment residuals, multi-view consistency) provide stronger or more stable supervision?
- The pipeline is expensive. Are there surrogate metrics, early stopping heuristics, or active-learning strategies that reduce the number of required generations for ranking?
- Could you present detailed failure examples and categorize them? Understanding failure modes in depth would be helpful for future work.

---

> ### Author Response · Authors · 2025-11-24
> **Official Comment by Authors**
>
> Thank you for the thoughtful review of our work! Below we address main questions and concerns:
>
> > **Dynamic degree decreases, showing that the method improves stability partly by reducing motion amplitude. There is a trade-off between temporal consistency and keeping lively, dynamic motion.**
>
> This is precisely why we evaluate whether the model still maintains strong capabilities of generating dynamic content, which is proven by other dynamic degree metrics and evaluation on in-the-wild dynamic scenes. Human Evaluation, mean SSIM and VLM-based metrics in Table 2, as well as performance on scenes with dynamic objects (Table 3) validate that the aligned model can generate various dynamic scenes.
>
> > **Although better than baseline, some videos still show flickering, blur, and occasional scene drift, especially in challenging or texture-heavy regions.**
>
> Yes, our claim is to improve 3D consistency, not to fully eliminate the issue. The main goal of the alignment is to maximize the likelihood of generating stable artifact-free camera trajectories. While the alignment generally improves the model quality, it is still restricted by the base model capabilities. The interesting observation is that a simple approach like ours can improve both static and dynamic videos.
>
> > **The dataset creation pipeline is computationally heavy and slow (as acknowledged in Appendix F), making the method difficult to scale or replicate.**
>
> The most time-consuming part of the pipeline is offline video generation. We agree that video generators are computationally expensive, yet even with a limited amount of resources, the method can be reproduced by utilizing a few-step distillation methods. We note that the whole pipeline fits on a 48Gb GPU, making it accessible and does not require a large cluster for video generation, as data can be generated in chunks or at different nodes. Moreover, we envision that our approach could be used during the training of video models such that post-hoc application will be unnecessary.
>
> > **Are there surrogate metrics, early stopping heuristics, or active-learning strategies that reduce the number of required generations for ranking?**
>
> We purposely generate more triplets than we use during training to ensure we obtain a sufficient amount of high-quality pairs with a significant gap in each pair so that the model can learn to better features that correspond to improved 3D consistency. In our experiments, three is the minimum amount of generated samples that allows for the extraction of pairs with a meaningful 3D consistency gap. Since epipolar error is simple and fast to compute, we can use it as an early stopping metric, skipping the generation of additional samples once the pair that satisfies the filtering criterion is found.
>
> > **Instead of relying only on pairwise epipolar distance, could multi-frame geometric reasoning (e.g., triangulation error, bundle-adjustment residuals, multi-view consistency) provide stronger or more stable supervision?**
>
> This is an interesting question! It is likely that other metrics will also work for the ranking and we explore some other options in Table 4. Here, we chose the Sampson epipolar error since it is robust and fast to compute.
> Moreover, not every metric will lead to improvements. For example, we show in the paper that the learned multi-view consistency metric MET3R produces too noisy scores for optimization. Bundle-adjustment residuals will likely be more accurate but are slow to compute.
>
> > **Could you present detailed failure examples and categorize them? Understanding failure modes in depth would be helpful for future work.**
>
> Thanks for the suggestion! The main failure case in the training pipeline is the false positives of the data filtering and scoring -- occasionally, some near-static scenes pass the threshold and are used in training. The aligned video generation model amplifies the 3D consistency of the base model but might still drift or produce artifacts in challenging scenarios with complex motions that are out of distribution for the base model. We have added the visual examples of the main fail cases to the updated anonymized webpage: https://anon-epipolar-dpo.github.io/anon-epipolar-dpo/

---

### Author Response · Authors · 2025-12-03
**Summary of Rebuttal and Discussion Phase**

We thank the reviewers (**QaX5**, **zotX**, **67U8**, **dEzy**) for insightful feedback and their effort to further improve the paper.

Our work targets the 3D consistency of video diffusion models by combining classical geometrical constraints with the preference alignment algorithm.
The novelty of our paper lies in:

1. Introducing a method for finetuning video diffusion models using epipolar geometry constraints as reward signals, particularly leveraging the Sampson distance to enhance 3D video consistency without needing differentiability. The models finetuned with simple yet reliable signals from classical computer vision algorithms achieve superior consistency and quality, significantly reducing artifacts and unstable motion trajectories in generated content.
2. Demonstrating that models trained on static scenes with dynamic cameras generalize well to dynamic scenes.
3. Building a large library of content useful for other methods and tasks.

### Rebuttal Summary

The key points of discussion were:

* **Comparison with a model trained directly on multi-view data (dEzy, zotX).** We agree this helps to better illustrate the effectiveness and necessity of our method. For the rebuttal, we fine-tuned Wan 2.1 with these datasets and observed that the aligned model quickly overfitted, consistently generating the same forward-facing trajectory regardless of the scene type, which limited the model's generalization. The analysis was added to the revised manuscript (Appendix B), and visual comparison is available on the updated project webpage.

* **Failure Case Analysis (QaX5)** We have added an expanded analysis of the failure cases both to the revised manuscript and project webpage. The main failure case in the training pipeline is the false positives of the data filtering and scoring. Occasionally, some near-static scenes pass the threshold and are used in training. The aligned video generation model amplifies the 3D consistency of the base model but might still drift or produce artifacts in challenging scenarios with complex motions that are out of distribution for the base model.

* **Computational Complexity (QaX5, dEzy, 67U8)** We acknowledge that generating an offline preference set for video diffusion models is expensive and non-trivial. Yet we note that this is the case for any video alignment algorithm, and this is the exact reason why we dedicated a lot of attention to parts such as prompt expansion (Section 3.3), ensuring we can generate videos with a meaningful 3D consistency gap even with a limited amount of resources. We will release the code and trained model for full reproducibility.

* **Handling Dynamic Objects (67U8, zotX, dEzy, QaX5)** All reviewers point out the effectiveness of the method, but highlight that the epipolar constraint is only valid for static scenes, and wonder how exactly we ensure that the model generalizes to dynamic content. This is exactly one of the main insights of our work: The alignment strategy allows us to train on static scenes with dynamic cameras where the epipolar metrics are robust and generalize well to dynamic scenes. This is enabled by a combination of implicit KL-divergence regularization with the reference model and static content penalty. Additionally, we hypothesize that the original base model has learned a disentangled representation for content and consistency, where motion is encoded very early (shown by works such as FreeInit that manipulate input noise to control the trajectory), allowing the adapter to generalize. We have clarified this in the revised manuscript and put the higher emphasis on this insight.


### Summary

During the rebuttal, we have further improved the evaluation by clearly demonstrating the core difference with direct finetuning on multi-view data, better emphasized the failure cases, and expanded the analysis of generalization to dynamic content. We believe these changes significantly improve the quality of the paper and resolve the reviewers' questions.
We highlight the overall positive review of dEzy, who explicitly mentions that they will improve the score if the concern of evaluation vs direct finetuning is added. We also want to emphasize that all reviewers had positive feedback about the idea, evaluation, and performance, but asked for additional clarification of generalization to dynamic content, which we agree is the most interesting insight of our work that required more emphasis, which was added to the revised manuscript.

---

### Meta-Review · Area_Chair_sEDG · 2025-12-24

**Summary:**

This paper proposes a method for enhancing 3D-consistent video generation in video models by constructing a dataset using the classical epipolar geometry framework and applying reinforcement learning with Flow-DPO. The method addresses the issue of 3D inconsistency commonly observed in existing video models. The authors design a complete experimental setup and metrics to demonstrate the effectiveness of their approach.
The reviewers acknowledge the innovation and effectiveness of combining traditional geometric methods with the new DPO-based reinforcement learning paradigm, and they recognize the strengths of the paper in experiments, dataset construction, and writing quality.

The reviewers primarily raised the following concerns:
- Since the training data are self-generated by the base model, the upper bound of performance is inherently limited by the capability of the base model itself. Even after optimization, some complex texture regions still exhibit flickering, blurring, or drifting, indicating that geometric consistency is not fully resolved. (zotX, QaX5, dEzy)
- Reviewers point out that constructing the offline preference dataset (around 54k generated videos) is extremely time-consuming and computationally heavy, posing challenges for the method’s scalability and reproducibility. (QaX5, 67U8, dEzy)
- Reviewers highlight that epipolar geometry is theoretically applicable only to static scenes, questioning how the method handles videos containing dynamic objects and how a model trained solely on static scenes can generalize effectively to dynamic scenarios. (67U8, zotX, dEzy, QaX5) In addition, reviewers observe that the method significantly reduces the dynamic degree (motion amplitude) of videos while improving geometric stability, seemingly avoiding errors by suppressing motion. They inquire whether this introduces a risk of over-penalizing motion. (QaX5, 67U8, zotX)
- Reviewers strongly recommend adding baselines where models are directly supervised-fine-tuned (SFT/LoRA) on multi-view or real datasets (e.g., DL3DV, RealEstate-10K) to demonstrate that performance gains stem from the proposed geometric constraints rather than simply from similarity between training and test distributions. They also question why real videos (ground truth) are not used as high-score samples in DPO. (dEzy, zotX)

In summary, this paper was reviewed by four experts in the field. The original recommendations were 2, 4, 4, 6. The reviewers like the novel idea of incorporating classical epipolar geometry constraints into video diffusion models, the comprehensive evaluation and the large-scale geometry preference dataset. The reviewers raised concerns on the lack of comparison against a direct SFT baseline on multi-view datasets, the applicability to dynamic objects/scenes, the trade-off between improved consistency and reduced motion, and the high computational cost of the offline data generation pipeline.
After the rebuttal, some of the concerns were adequately addressed. However, considering the large number of existing geometry-constrained methods in the current literature, the comparisons provided in this work remain insufficient. Furthermore, the performance gains over prior methods are relatively marginal. Although the method shows a certain degree of generality, epipolar geometry framework is still theoretically limited to the static scenes.

**Reviewer Concerns:**

**Well addressed:**
- Regarding the concern that the method is limited by the capability of the base model, and why the authors do not directly train on external datasets (zotX, dEzy): The authors have theoretically analyzed that reinforcement learning essentially optimizes distributions, and using external datasets may introduce distributional shift. Therefore, they use videos generated by the video base model itself as the preference dataset. As a reinforcement learning paper, this is reasonable, since this is a common setup in DPO.
- Failure case analysis (zotX, dEzy): The authors have added detailed failure-case analyses on both the project page and in the paper.
- Generalization to dynamic scenes (67U8, zotX, dEzy, QaX5): The authors clearly explain why training on static scenes can generalize to dynamic ones. They argue that, through implicit KL-divergence regularization and the lightweight LoRA adapter (only 2.5% of the model weights), the model learns to decouple motion control from content generation. Thus, even though the reward is obtained only from static scenes, the learned geometric stability can transfer to dynamic scenarios. The authors emphasize that the evaluation results in Table 3 and VBench support this point. This is reasonable.
- Lack of key baseline comparison (SFT vs. DPO) (dEzy, zotX): The authors conducted the additional experiments strongly requested by reviewer dEzy, performing supervised fine-tuning (SFT) with LoRA on the DL3DV and Real10K datasets using the Wan 2.1 model.

**Partly addressed:**
- Computational cost and dataset construction issues (QaX5, 67U8, dEzy): The authors acknowledge that offline video generation is very time-consuming, but claim that techniques such as distillation can help accelerate the process. They also state that the entire pipeline can run on a single 48GB GPU (reducing hardware requirements). This partially alleviates concerns about reproducibility. QaX5 acknowledges the effort and positive contribution of constructing the dataset.
- Trade-off between motion amplitude and consistency (QaX5, 67U8): The authors explain that some degree of motion suppression is necessary to eliminate geometric hallucinations and drifting. Although the authors analyze the influence of the parameter $\lambda$, they do not propose solutions that can simultaneously maintain high motion dynamics and high geometric consistency.

**Unsolved:**

The major concerns have all been resolved.

**Reviewer Scores:**

**dEzy (2):**

Although the reviewer initially gave the lowest rating, they clearly stated conditions for raising their score (``If you fix this problem, I will raise my score ``), all of which the authors have now satisfied. Thus, dEzy would likely raise the score to 4-6.

**QaX5 (4):**

The reviewer initially indicated they ``wouldn’t mind accepting`` the paper, and their concerns have since been addressed. With the added failure analysis and a transparent discussion of limitations, the work now appears more solid. However, the epipolar geometry framework is still theoretically limited to the static scenes. Thus, QaX5 would likely keep the score.

**zotX (4):**

The reviewer’s main concerns have all been addressed. Since the central question regarding real data received strong theoretical and empirical support, zotX would slightly raise the score.

**67U8 (6):**

67U8's concerns are centered on the handling of dynamic objects. Given that this reviewer was leaning toward acceptance and the authors provided clear, supportive explanations (which however not fully addressed), the score is expected to remain at 6.

---

### Decision · Program_Chairs · 2026-01-26

Reject